# Continual Learning of a Transformer-Based Deep Learning Classifier Using an Initial Model from Action Observation EEG Data to Online Motor Imagery Classification

**DOI:** 10.3390/bioengineering10020186

**Published:** 2023-02-01

**Authors:** Po-Lei Lee, Sheng-Hao Chen, Tzu-Chien Chang, Wei-Kung Lee, Hao-Teng Hsu, Hsiao-Huang Chang

**Affiliations:** 1Department of Electrical Engineering, National Central University, Taoyuan 320, Taiwan; 2Pervasive Artificial Intelligence Research Labs, Hsinchu 300, Taiwan; 3Department of Rehabilitation, Taoyuan General Hospital, Taoyuan 330, Taiwan; 4Division of Cardiovascular Surgery, Department of Surgery, Taipei Veterans General Hospital, Taipei 112, Taiwan; 5Department of Surgery, School of Medicine, College of Medicine, Taipei Medical University, Taipei 110, Taiwan

**Keywords:** brain computer interface, electroencephalography (EEG), action observation, motor imagery, transformer network

## Abstract

The motor imagery (MI)-based brain computer interface (BCI) is an intuitive interface that enables users to communicate with external environments through their minds. However, current MI-BCI systems ask naïve subjects to perform unfamiliar MI tasks with simple textual instruction or a visual/auditory cue. The unclear instruction for MI execution not only results in large inter-subject variability in the measured EEG patterns but also causes the difficulty of grouping cross-subject data for big-data training. In this study, we designed an BCI training method in a virtual reality (VR) environment. Subjects wore a head-mounted device (HMD) and executed action observation (AO) concurrently with MI (i.e., AO + MI) in VR environments. EEG signals recorded in AO + MI task were used to train an initial model, and the initial model was continually improved by the provision of EEG data in the following BCI training sessions. We recruited five healthy subjects, and each subject was requested to participate in three kinds of tasks, including an AO + MI task, an MI task, and the task of MI with visual feedback (MI-FB) three times. This study adopted a transformer- based spatial-temporal network (TSTN) to decode the user’s MI intentions. In contrast to other convolutional neural network (CNN) or recurrent neural network (RNN) approaches, the TSTN extracts spatial and temporal features, and applies attention mechanisms along spatial and temporal dimensions to perceive the global dependencies. The mean detection accuracies of TSTN were 0.63, 0.68, 0.75, and 0.77 in the MI, first MI-FB, second MI-FB, and third MI-FB sessions, respectively. This study demonstrated the AO + MI gave an easier way for subjects to conform their imagery actions, and the BCI performance was improved with the continual learning of the MI-FB training process.

## 1. Introduction

Brain computer interface (BCI) is the technology for people to directly communicate with external devices through the acquisitions and translations of their brain activities. A BCI system measures the neural activities from the central nervous system (CNS) and converts them into artificial outputs in order to replace, restore, enhance, supplement, or improve natural CNS output [1]. Several BCI systems have been developed based on different brain imaging modalities, such as electroencephalography (EEG), magnetoencephalography (MEG), functional magnetic resonance imaging (fMRI), and functional near-infrared spectroscopy (fNIRS) [2]. In contrast to the fMRI and fNIRS which measure the slow changes of hemodynamic responses, the EEG/MEG records fast changes of electrophysiological signals with a high sampling rate and enables the possibility of timely observation for the neural activities inside the brain. In particular, the EEG has the advantages of easy preparation, inexpensive equipment cost, and high temporal resolution. It has been chosen for a wide variety of clinical applications, such as sleep disorder diagnosis [3], seizure detection [4], emotion classification [5], etc. Advanced signal processing techniques for EEG have also been developed to avoid the interference of external artifacts (e.g., electromyography, motion artifact, etc.) [6,7]. Owing to the aforementioned reasons, EEG is the most popular choice to implement a BCI system [7].

An EEG-based BCI system requires an elaborately designed task for generating reliable neural responses, a credible signal processing method for extracting particular signal features, and an efficient translation algorithm to produce output signals [8]. Current BCIs can be categorized into endogenous and exogeneous BCIs, according to the necessity of an external stimulus for generating brain activities. Endogenous BCIs utilize spontaneously generated brain patterns (e.g., motor imagery [9], speech imagery [10], slow cortical potential [11], etc.), whereas exogeneous BCIs detect brain patterns induced by external stimuli (e.g., steady-state visual evoked potential [12], flash visual evoked potential [13], steady-state auditory response [14], etc.). Though the exogeneous BCIs have the benefits of less training and a higher information transfer rate (ITR), the indispensable external stimuli, such as flickering LEDs or auditory beeps, are required to evoke discriminative patterns. Patients with locked-in syndrome (LIS) usually have weak muscle activities as well as deficits in their sensations [15], so that limits the feasibility of exogeneous BCI in severe disabilities. On the other hand, endogenous BCIs are independent of external stimuli which utilize neural signals generated from designated mental imagery tasks. Two BCIs, the motor imagery (MI) and speech imagery BCIs, are treated as the two most promising systems because they are operated in more natural ways of our daily life. In contrast to the speech imagery BCI, the motor imagery BCI has been most studied, owing to the cross-individual generality in its task execution and the abundant data resources in the previous literature. For well-trained users, the MI-based BCI system, denoted as the MI-BCI system, can be a straightforward, convenient, and fast way to communicate with external devices.

The MI-BCI requests users to perform extensive training of imagination actions before they are capable of generating desired brain patterns to achieve acceptance performances. Traditional MI-BCIs requested users to imagine the movements of their limbs (e.g., legs or hands) in response to a visual cue presented on a screen with or without feedback [16]. Khare et al. (2022) developed an automatically tuning algorithm to optimize the selection of tunable Q wavelet transform (TQWT) parameters and a high detection accuracy was achieved [17]. Khare et al. (2020) utilized flexible variational mode decomposition (F-VMD) to extract meaningful EEG components. They calculated hjorth, entropy, and quartile features from F-VMD decomposed components and a flexible extreme learning machine (F-ELM) was used to classify the EEG features from different MI tasks [18]. Filho et al. (2022) studied the event-related desynchronization (ERD) occurrence and classification accuracy under the conditions of no feedback and actual feedback in MI tasks. With feedback, they found not only the detection accuracies but also the ERD values in both contralateral and ipsilateral sensorimotor cortexes were significantly improved [19]. Alimardani et al. (2018) discussed the training effects of MI-BCI with different visual feedback representations. They concluded that humanlike visual feedback (e.g., human hand action) achieved a better MI learning performance, compared to non-humanlike visual feedback (e.g., robot gripper or direction bar) [20]. Frideman et al. (2008) studied the performance of a visual-feedback MI-BCI in a virtual reality (VR) environment. They found that a highly-immersive visual environment can strengthen the subjective impression and result in a better training performance [21]. In addition to visual feedback, other BCI systems are designed to train user’s MI responses with different feedback modalities, such as auditory or tactile feedbacks. McCreadie et al. (2013) studied the feasibility of replacing visual feedback with stereo auditory feedback [22]. They demonstrated that stereophonic feedback could effectively achieve comparable results with the use of visual feedback. Nijboer et al. (2008) compared the performances of auditory feedback training with visual feedback training in an MI-BCI system [23]. No difference in the BCI performances was found between the uses of the two feedback modalities. Ishihara et al. (2020) reviewed the effectiveness of feedback modalities in 184 BCI studies [24], in which the visual feedback was the most effective approach for neurofeedback in MI-BCI training. Owing to the advantages of an intuitively, highly immersive sensation and a better flexibility for experimental design, the visual feedback is most widely chosen to implement a BCI system.

Though visual feedback has been demonstrated as an effective and efficient way for neural training in MI-BCI system [25], user’s training performance is highly-related to the immersive feeling and self-perception [26]. Ono et al. (2013) compared the values of event-related desynchronization (ERD) in the conditions of no feedback, bar feedback, and incongruent/congruent feedback. In the incongruent and congruent feedbacks, a real hand open/grasp picture was displayed on a screen. They found the participants in a congruent condition achieved higher ERD values and better detection accuracies. They also demonstrated the visual feedback of real-hand picture-induced higher ERD values than bar feedback or no feedback. Achanccaray et al. (2018) designed an MI-based BCI system to control a 3D arm in a VR environment [27]. Ziadeh et al. (2021) [26] studied the influence of embodiment in the VR environment on a subject’s BCI performance. They found both the senses of the ownership and agency can influence the MI training performance [28].

Most MI-BCI systems request subjects to participate in multiple experiment sessions before they are able to operate the MI-BCI with an acceptable detection rate, including MI sessions and MI-feedback (MI-FB) sessions [26,29]. In the MI-FB sessions, participants performed the same MI tasks but received online classification results via feedbacks. However, previous research has indicated that a nonnegligible portion of subjects (about 15~30%) were not able to control the BCI systems through the aforementioned BCI training process [30]. Vidaurre et al. (2011) hypothesized two reasons for the problems related to the successful operation of an MI-BCI system, in which the first reason is the lack of a generic pattern for initial BCI training, and the second reason is the difficulty of transition from offline calibration to online feedback. For the first point, the creation of a generic pattern for a classifier setup is difficult since most BCI training lacks a clear instruction for movement imagination which could result in large inter-subject difference in the induced brain wave patterns. It has also been difficult to obtain a consistent brain wave pattern in previous studies due to the diversity of MI task design in different BCI systems. For example, some studies requested participants to imagine the simple moving of an object on the screen [22], while the others claimed a more complicated MI task, such as imagining a serial movement of a hand or a foot. Recent studies have also demonstrated that the training efficacies can be enhanced by creating the illusions of ownership and agency [20,28,30] which can be conveniently achieved by wearing a VR head-mount device (HMD) [21,26,27,31,32]. For the second point, most BCI studies usually created their initial classifiers from the EEG data in MI tasks with no feedback, and tried to modify the initial classifier to fit online MI-FB applications [33]. Nevertheless, the gap between the initial model and the final classifier could be huge, so that subjects usually had to participate in a large amount of MI-FB training until acceptable results were obtained.

In this study, we intend to study the feasibility of building the initial classifier for a BCI system using the EEG data obtained from an action observation (AO) task. Action observation has been proposed as an effective rehabilitation therapy approach based on the role of a mirror neuron system in motor learning [34]. The mirror neuron system is activated during the observation of an action execution. In contrast to mirror therapy (MT) which requests semi-paralyzed patients to activate the mirror neuron system by watching the reflection image of moving unaffected sides in a mirror [35], the AO therapy conducts the formation of motor memory by viewing the limb movements which can provide a more flexible way in experimental design. Hsieh et al. (2020) compared the rehabilitation outcomes between the interventions of AO therapy and MT in stroke patients [36], and they concluded the AO therapy can lead to better improvements in patients. Vogt et al. (2013) reviewed neuroimaging studies and claimed neural activities in motor cortices can be enhanced by performing concurrent AO and MI [37]. Though the role of AO in motor activities has been studied in the previous literature [27,38,39], most BCI studies utilized AO as a feedback approach for improving a subject’s MI performance rather than using the brain activities in AO for constructing an initial BCI classifier. Since the poor setting of initial BCI classifiers could frustrate the subject and impede training efficiency [40], the AO, whose brain activation areas are located in conjunction with the brain regions recruited in an MI task [38], could be a possible way to generate similar signal features for the initial setup of a BCI classifier. This study aims to initiate the parameter settings of BCI classifiers based on the collection of EEG data from AO plus the MI (AO + MI) task. We discuss the feasibility of tuning this initial classifier to achieve better classification results by means of using continual learning with MI-FB data. The AO + MI provides a convenient way for subjects to follow the experimental designer’s instruction. It also gives an easier way for subjects to conform their imagery actions with the expected task design.

The present study intended to build a training method for MI-BCI based on AO + MI in the VR environment. We aimed to answer the following questions: (1) Is the classifier trained from AO + MI data good enough for MI classifications? (2) Can we generate a classifier from AO + MI data and tune the classifier with the continual learning of MI data? (3) Can we adopt a transformer-based deep learning classifier in our BCI training? (4) Can the detection accuracy of the classifier which is trained from cross-subject AO + MI data be improved by the continual learning of individual MI data? Because both the AO and MI are two motor tasks with no actual motor executions but sharing overlapped motor areas in their activations, we wonder if the requisite of large amount of EEG data in tedious MI task could be replaced by EEG data from AO + MI task in an easier way.

## 2. Materials and Methods

### 2.1. Subjects and EEG Recordings

Five normal subjects (5 males, all right-handed subjects; mean age = 24 ± 4.2 years) were recruited to participate in our study. All participants were requested to sit in a comfortable armchair in a dimly illuminated electro-magnetic shielded room. EEG signals were recorded using an eight-channel dry-electrode wireless EEG system (bandpass 0.05–250 Hz; 55–65 Hz bandstop; 24-bits data resolution; digitized at 1 kHz; InMex EEG system, WellFulfill Co., Taoyuan, Taiwan). The EEG electrodes were placed in accordance with the international 10–20 system. Electrodes were located at FC1, FC2, C3, Cz, C4, P1, and P2 positions, with respect to a reference electrode placed at the right mastoid and a ground electrode placed at the left mastoid. Each EEG dry electrode was constructed by 10 spring-loaded copper pins, and its biocompatibility was certificated by ISO 10993. Immersive scenarios were created in the virtual reality environment using Unity 3D Engine (Unity Technologies Inc., San Francisco, CA, USA) and provided to the participants by wearing a Head Mounted Display system (HTC VIVE, HTC Co., Taipei, Taiwan) (see Figure 1).

### 2.2. Experimental Task

Participants came to participate in our experiment for four weeks, one time in each week. In the first week, subjects were requested to participate in an AO + MI session, and then they were asked to join three MI-FB sessions in the following three weeks. The VR environment was programmed using Unity 3D and displayed on a head-mount display (HMD) device to provide an immersive experience for the subjects. Subjects were asked to imagine they were looking into a mirror and the virtual character was the reflection image of themselves. In the AO + MI session, each trial contained two AO + MI time blocks and one MI time block (without feedback). In each trial, the designated hand, either the left or right hand, was chosen randomly. During the AO + MI time block, the virtual character raised its designated hand once, and subjects were requested to perform an MI movement of the same hand concurrently with the action of the virtual character. During the MI time block, subjects were asked to perform an MI movement of the same designated hand just after they saw the presence of the direction sign (see Figure 1). Subjects were requested to perform 300 trials in the AO + MI session, 150 trials for left hands and 150 trials for right hands, arranged in a random order. For every twenty trials, subjects had a five-minute break to prevent them from mental fatigue. Each AO + MI time block was the concatenation of a resting period (5~7 s) and an AO + MI period (2 s). In the resting period of the AO + MI time block, the virtual character was kept still with no movement. In the AO + MI period, the virtual character performed a front arm raise of the designated hand once. Subjects were instructed to imagine the virtual character was the reflection images of themselves, similar to looking into a mirror. In the MI time block, subjects were requested to perform a mental rehearsal of the arm raise action which they viewed in the previous AO + MI time block. One resting period (5~7 s) was concatenated with an MI period (2 s). In the resting period of the MI time block, a cross sign was presented, and subjects were asked to relax without movement intention. In the MI period, a direction sign was presented, and subjects were requested to perform a mental rehearsal of the arm raise action of the designed arm which they viewed in the AO + MI time block.

After the AO + MI sessions, subjects were requested to participate in three MI-FB sessions in the following three weeks, one session in a week. Each MI-FB session contained 150 trials of MI movements (with visual feedback), one-third of the trials (50 trials) for left-hand imagery movements, one-third of the trials (50 trials) for right-hand imagery movements, and one-third of the trials (50 trials) for no movement, arranged in a random order. In each trial, a preparation sign was presented before the movement indication sign. Subjects were initially instructed to keep themselves relaxed in a preparation time block (5~7 s), and then a left or right direction sign was presented to instruct subjects which hand should be performed in the MI movement time block (3 s). For each MI movement, subjects were requested to perform a mental rehearsal of the arm raise action of the designated hand that they viewed in the AO + MI session. The EEG signals in each MI time block were classified into one of the three classes (i.e., the left-arm movement, the right-arm movement, and the resting state), and the classification result was feedbacked to the subject by driving the virtual character to show the arm raise movement of the classified result in the feedback time block (2 s), accompanied with a circle or cross sign to respond to the correctness of the MI classification in this trial. The experimental paradigms of the AO + MI and the MI-FB sessions are shown in Figure 2a and Figure 2b, respectively. All participants gave informed consent, and the study was approved by the Ethics Committee of the Institutional Review Board (IRB) (TYGH 107055), Tao-Yuan General Hospital, Taiwan. All measurements were noninvasive, and the subjects were free to withdraw at any time.

### 2.3. Transformer-Baed Spatial-Temporal Network (TSTN) for MI Classification

In this study, we adopted the work proposed by Song et al. (2021) [41], which applied attention mechanisms on the spatial and temporal features of EEG data. The eight-channel EEG signals were prefiltered within 4~40 Hz (3rd-order Butterworth IIR filter) and then downsampled to 250 Hz. The EEG data were then segmented into two-second epochs and spatially filtered using a common spatial pattern (CSP) to extract the discriminant features. The spatial features in the feature-channel data were further enhanced by applying spatial transforming with an attention mechanism. The enhanced feature-channel data were segmented across time and segmented into embedded patches using two convolution layers. The relationship among different temporal patches was further perceived using multi-head transforming to obtain distinguishable representations. The features processed by the temporal transformer were averaged and a fully connected layer was used to classify the feature-segment representations into one of the three categories (i.e., the left-arm movement, the right-arm movement, and the resting state). The detailed signal processing is described as follows.

#### 2.3.1. Spatial Filtering Using Common Spatial Pattern (CSP)

CSP is an effective feature extraction method which maximizes the discriminability of two classes by constructing a set of optimized spatial filters [41]. In our study, the CSP was constructed based on a one-versus-rest (OVR) strategy, i.e., rest vs. left/right, left vs. rest/right and right vs. rest/left), to extract feature-channel signals. For each CSP, the covariance metrices R_1_ and R_2_ were calculated from two sets of EEG signals X_1_ and X_2_, in which C is the number of EEG channel (C = 8), *T* is the length of sampled data (*T* = 2000), and $X_{1}\in {ℜ}^{C\times T}$ and $X_{2}\in {ℜ}^{C\times T}$ are the EEG data of the classes that we want to identify. The CSP seeks to find a matrix that contains eigenvectors $P=\left[\begin{array}{cccc} p_{1} & p_{2} & \ldots & p_{c} \end{array}\right]$ for simultaneously diagonalizing the covariance matrixes R_1_ and R_2_, which can be represented as:(1)$$D=P^{T}R_{1}P$$

And
(2)$$I=P^{T}R_{2}P$$
in which D is the diagonal matrix with eigen values {λ_1_, λ_2_, …, λ_c_} sorted in descending order and I is the identify matrix. The first two columns *p*_1_ and *p*_2_ in P were chosen, and the transpose of the first two column vectors $p_{1}^{T}$ and $p_{2}^{T}$ in the subfilters of the three CSPs were stacked as the spatial filter Z $\left(Z\in {ℜ}^{6\times c}\right)$. For an EEG data matrix X ($X\in {ℜ}^{c\times T}$), the filtered EEG data S can be represented as:(3)S=ZX
where $S\in {ℜ}^{6\times T}$ contains the feature-channel signals filtered by the six subfilters obtained from the three discriminations, with two subfilters for each discrimination.

#### 2.3.2. Spatial Transforming for the Enhancement of Feature-Channel Signals

The feature information in the feature-channel data S was enhanced by applying the self-attention mechanism. The input vectors were multiplied with three weighted matrices *W_q_*, *W_k_*, and *W_v_* to obtain the key, query, and the value vectors for each of the input vectors, represented as:(4)$$q_{i}=W_{q}a_{i}$$
(5)$$k_{i}=W_{k}a_{i}$$
and
(6)$$v_{i}=W_{v}a_{i},$$
in which *a*_i_ is the *i*th input vector, and *q*_i_, *k*_i_, and *v*_i_ are the query, key, and value vectors. The similarity between *q*_i_ and *k*_j_ is then estimated by calculating the scaled dot-product:(7)$$S(q_{i},k_{j})=\left(q_{i}\cdot k_{j}\right)/\sqrt{d_{k}}$$
where *d* (*d* = 500) is the dimension of k_j_.

The value of S(*q_i_*, *k_j_*) is then normalized using the softmax function to obtain ${\hat{\alpha }}_{i,j}$ and then multiplied by vector *v_j_* to obtain the *i*th output vector *b_i_*. The calculation can be represented as the following equation:(8)$$b_{i}=\sum_{j}\mathrm{Softmax}\left(\frac{q_{i}\cdot k_{j}}{\sqrt{d_{k}}}\right)\cdot v_{j}=\sum_{j}{\hat{\alpha }}_{i,j}v_{j}$$

The whole process can be represented as:(9)$$B=\mathrm{Attention}(Q,K,V)=\mathrm{Softmax}\left(\frac{QK^{T}}{\sqrt{d_{k}}}\right)V$$
where the output matrix B contains the enhanced feature-channel data.

#### 2.3.3. Patch Embedding of Feature-Channel Signals

In order to reduce computation complexity, the feature-channel data were segmented into data patches and the dependencies among different patches in a trial were learned using a temporal transforming network (see below). The enhanced feature-channel data B were passed through a one-dimension convolution layer (number of filter = 2; kernel size = 51) and then processed by a two-dimension convolution layer (number of filter = 10; kernel size = 6 × 5) to creat 10 signal patches, in which each patch has a data length of 90 samples. The 10 patches were used as inputs for the following temporal transforming network.

#### 2.3.4. Temporal Transforming for Embedded Patches

Multi-head attention (MHA) was employed to explore the dependencies among different patches in our temporal transforming step. The MHA allows the network to attend parts of the sequence differently. The embedded patches (10 signal patches) were divided into five data sets E_i_ ($E_{i}\in {ℜ}^{2\times 90};i=1,\cdots ,5)$ passed through *h* (*h* = 5) independent attention networks. The outputs of these attention networks were concatenated and combined together with a final weight matrix $W^{o}$. The MHA can be represented as follows:(10)$$F=\mathrm{Multihead}\left(Q,K,V\right)=\mathrm{Concat}\left({\mathrm{head}}_{1},{\mathrm{head}}_{2},\cdots ,{\mathrm{head}}_{h}\right)W^{o}$$
where F $\left(F\in {ℜ}^{10\times 90}\right)$ is the output of multihead attention, ${\mathrm{head}}_{i}=\mathrm{Attention}\left(Q_{i},K_{i},V_{i}\right)$ is the output of *i*th head, Concat (. ) is the function to concatenate the output of each head together, $Q_{i}=W_{i}^{Q}E_{i}$, $K_{i}=W_{i}^{K}E_{i}$, $V_{i}=W_{i}^{V}E_{i}$, and $\left(Q_{i},K_{i},V_{i}\right)\in {ℜ}^{2\times 90}$, $\left(W_{i}^{Q},W_{i}^{K},W_{i}^{V}\right)\in {ℜ}^{2\times 2}$ and $W^{o}\in {ℜ}^{10\times 10}$.

#### 2.3.5. Classifier

The output of multi-head attention *F* was then passed to an averaging pooling layer, normalized, and then the classification of imagery movements was achieved by a fully-connected layer, in which cross-entropy was chosen as its loss function. The architecture of the TSTN network is shown in Figure 3.

### 2.4. Training of the TSTN Classifier

It has been demonstrated that both the AO and the MI have significant effects on the modulation of sensorimotor Mu rhythms [42]. The clinical literature in rehabilitation studies found that the AO + MI is an effective way to restore motor function in stroke patients [38]. Since AO and MI activate overlapped motor-related brain areas, it is reasonable to construct an initial classifier based on AO + MI data and then use MI-feedback data to continually train the network for better MI classification results.

Figure 4 shows the training and the testing procedure of our TSTN classifier. The classifier was initially trained using all the AO + MI data (900 trials for each subject; 300 trials for each class). The classifier obtained from the AO + MI data, denoted as TSTN_AO+MI_, was used to test the classification performance in the MI (without feedback) task. Because inaccurate trials in MI training could impede the user’s BCI performance [43], only those trials in MI data (without feedback) which were correctly identified were chosen for the continual learning to obtain TSTN_MI_. The TSTN_MI_ was then applied to test the classification performance in the 1st MI-FB data (with VR feedback), and the correctly classified trials were collected to train a new classifier TSTN_MI-FB_1_. The TSTN_MI-FB_1_ was again used to test the classification performance in the 2nd MI-FB data (with VR feedback), and the correctly classified trials were used to train a final classifier TSTN_MI-FB_2_. The final TSTN_MI-FB_2_ was applied to test the BCI performance in the 3rd MI-FB training dataset and the test performances among different steps were compared to see how the BCI training improved the classifier performances. For the model training, Adam with a learning rate of 0.0002 was utilized and batch size was set as 50. The dropout rate was 0.3 in spatial transforming and 0.5 in temporal transforming [40,44]. Ten-fold cross validation was applied to evaluate the final results, with each fit being performed on a training set consisting of 90% of the total training set selected at random, with the remaining 10% used as a hold out set for validation.

### 2.5. Comparing the Detection Performance with Other Classifiers

In order to compare the effectiveness of the TSTN with other classifiers, two convolutional neural network (CNN) based classifiers, the EEGNet [45] and the DeepConvNet [46], and three support vector machines (SVMs) with kernel functions of linear, radial basis, and polynomial functions, were tested and compared with TSTN. The training and testing procedures were the same as our training procedures shown in Figure 4.

## 3. Results

In this study, we built the initial classifier TSTN_AO+MI_ using AO + MI data, and then performed continual learning with chosen trials in MI, first MI-FB, and second MI-FB datasets to obtain TSTN_MI_, TSTN_MI-FB_1_, and TSTN_MI-FB_2_, respectively.

Table 1 shows the classification results of using TSTN in our five participants. The test targets for the TSTN_AO+MI_, TSTN_MI_, TSTN_MI-FB_1_, and TSTN_MI-FB_2_ were the MI (without feedback), first MI-FB, second MI-FB, and third MI-FB datasets, respectively. The classification accuracies were increased after each continual learning step, in which the classification accuracies averaged over the five participants were 0.63, 0.68, 0.75, and 0.77 for the TSTN_AO+MI_, TSTN_MI_, TSTN_MI-FB_1_, and TSTN_MI-FB_2_, respectively. The specificities were 0.81, 0.84, 0.87, and 0.89 for the TSTN_AO+MI_, TSTN_MI_, TSTN_MI-FB_1_, and TSTN_MI-FB_2_, respectively. The F1 scores averaged over the five subjects were 0.63, 0.68, 0.75, and 0.77, respectively. It is worth noting that the initial TSTN_AO+MI_ had already achieved an accuracy higher than 60% and the detection accuracies kept increasing with the classifier tuning in each continual learning step. This demonstrated the feasibility of using AO + MI data to create an initial classifier model for MI classification.

Figure 5 plots the detected accuracies of the three MI classes in the five subjects by applying the TSTN_AO+MI_, TSTN_MI_, TSTN_MI-FB_1_, and TSTN_MI-FB_2_ to the MI (without feedback), first MI-FB, second MI-FB, and 3rd MI-FB datasets, respectively. The detected accuracies obtained from S1 to S5 are shown in Figure 5a–e, and the averages of detection accuracies are shown in Figure 5f. The detected accuracies of individual TSTN_AO+MI_ on MI dataset in each subject were 0.74, 0.58, 0.61, 0.61, and 0.60 for S1, S2, S3, S4, and S5, respectively. For individual TSTN_MI_ on the first MI-FB data in each subject, the detected accuracies were 0.73, 0.66, 0.62, 0.65, and 0.73 for S1, S2, S3, S4, and S5, respectively. For individual TSTN_BCI_1_ on the second MI-FB data in each subject, the detected accuracies were 0.78, 0.72, 0.73, 0.73, and 0.78 for S1, S2, S3, S4, and S5, respectively. For individual TSTN_MI-FB_2_ on the third MI-FB data in each subject, the detected accuracies were 0.83, 0.73, 0.75, 0.73, and 0.80 for S1, S2, S3, S4, and S5, respectively. The averages of the detection accuracies over the five subjects are shown in Figure 5f.

In addition to the model built from individual data, we were interested in testing whether the data collected from different individuals could be pooled to train a BCI classifier. The TSTN_AO+MI_all_, TSTN_MI_all_, TSTN_MI-FB_1_all_, and the TSTN_MI-FB_2_all_ were obtained, following the training procedure described in Figure 4, while the measured data over the five subjects were pooled for classifier training. The classification accuracies, specificities, and F1 scores in the five subjects are listed in Table 2. The averaged accuracies in the five subjects were 0.61, 0.63, 0.68, and 0.70 for the TSTN_AO+MI_all_, TSTN_MI_all_, TSTN_MI-FB_1_all_, and the TSTN_MI-FB_2_all_, respectively. The averaged specificities in the five subjects were 0.78, 0.82, 0.84, and 0.84 for the TSTN_AO+MI_all_, TSTN_MI_all_, TSTN_MI-FB_1_all_, and the TSTN_MI-FB_2_all_, respectively. The averaged F1 scores in the five subjects were 0.61, 0.64, 0.67, and 0.70, respectively.

Figure 6a–e presents the detection accuracies using the classifiers trained from the pooled data of the five subjects. The averaged accuracies over the five subjects are shown in Figure 6f. It can be observed that the detection accuracies were increased with the continual training, which might indicate the feasibility of pooling subjects’ data to obtain a generally applicable model. The detected accuracies of TSTN_AO+MI_all_ for MI data were 0.65, 0.59, 0.62, 0.63, and 0.58 for S1, S2, S3, S4, and S5, respectively. For TSTN_MI_all_, the detected accuracies for the first MI-FB data were 0.66, 0.65, 0.62, 0.63, and 0.61 for S1, S2, S3, S4, and S5, respectively. For TSTN_MI-FB_1_all_, the detected accuracies for the second MI-FB data were 0.71, 0.65, 0.70, 0.68, and 0.65 for S1, S2, S3, S4, and S5, respectively. For TSTN_MI-FB_2_all_, the detected accuracies for the third MI-FB data were 0.73, 0.67, 0.71, 0.69, and 0.68 for S1, S2, S3, S4, and S5, respectively.

To demonstrate the effectiveness of the TSTN network in this study, the EEGNet [45], DeepConvNet [46], and three SVMs with kernel functions of linear, polynomial, and radial basis functions were compared. All the classifiers were built according to the training procedure listed in Figure 4. The detected accuracies obtained from the TSTN network were compared with the five classifiers and the detection accuracies are listed in Table 3. It can be observed that all the classifiers showed increased detection accuracies in our continual learning process. The three deep learning networks (i.e., the TSTN, EEGNet, and DeepConvNet) showed comparable detection accuracies in the third MI-FB data, which were 0.77, 0.74, and 0.75 for TSTN, EEGNet, and DeepConvNet, respectively. For the initial model, the TSTN_AO+MI_ had the highest detection accuracy (0.63) in the detection of MI data, compared to other classifiers. This might have been due to the substantial differences between the AO + MI and MI/MI-FB datasets. The AO + MI task activated both the neural circuitries of action observation and motor imagery, while the MI/MI-FB activated the neural circuitry of motor imagery only. Because the transformer has the greater ability to model long-distance dependencies among temporal embedded patches, it could provide more possibilities to find the dependency between different datasets.

## 4. Discussion

This study aimed to study the feasibility of using EEG data induced from AO + MI to build an initial model for MI-BCI, and the model could be continually improved with the provision of MI data in continual learning. Compared to the MI training in previous BCIs [47], those BCIs requested subjects to perform imagery movements by providing a simple visual or auditory cue. The lack of clear instructions makes it difficult to conform the ways for subjects to perform imagery movements. Therefore, if we can give subjects clear instructions to perform imagery movements, we might have the chance to reduce inter-individual difference and obtain generic brain wave patterns for training BCI classifiers. Alimardani et al. (2022) surveyed the impacts of different demographic and psychological variables [48] on the performance of imagery movement BCI in 54 subjects. They suggested the needs of prior BCI training and clear instructions for the experimental protocol are two of the most important factors which can influence a subject’s BCI performance. In our previous study [49], the lack of explicit MI instructions would make it difficult for subjects to follow. Subjects may try to develop their own strategies to achieve better BCI accuracies. The distinct strategies in imagery movements could result in large inter-individual variations in the induced brain wave patterns, which makes the group analysis of MI data difficult.

In our research, we used the data obtained from the AO + MI task to create an initial model for MI-BCI control. The AO + MI task requested users to passively watch the action of the characters in the VR environment and concurrently imagine themselves performing the same movements. According to [50], Cengiz et al. (2018) studied ERD in the sensorimotor Mu rhythm when the subjects were performing an AO task. Zhanget et al. (2018) also found that AO can promote the effect of motor relearning [51], and the AO task can be performed concurrently with MI (AO + MI) to achieve more effective motor learning or a rehabilitation setting [38]. In our experiment, we asked the subjects to perform imagery movements the same as the actions of the virtual character, in order to achieve a better motor learning effect. It has been reported in the previous literature that the task of AO + MI can be performed either from the first person visual perspective [52] or from the third-person visual perspective [53,54]. Some studies have claimed the motor activities in AO + MI from the third-person visual perspective might involve rotation actions to transform the third-person visual perspective into a first-person imagery perspective [54]. However, it has been reported that the rotation and the transforming effects can be removed with the provision of a clear instruction, by asking subjects to imagine that the observed movements in the third-person perspective is similar to viewing the images in a mirror [38]. In our study, we instructed subjects to imagine that the virtual character is the reflection image of themselves in a mirror. The imagery actions in the following MI and MI-feedback tasks were the same as the actions they viewed in the AO task. Therefore, subjects recalled the images of virtual characters’ actions in their MI and MI-FB tasks. Since the imagery movements in the AO + MI, MI, and MI-FB tasks were the same, this could be the reason why the detection accuracy is able to be improved in the continual learning process.

In this paper, we adopted the transformer-based deep learning network proposed by Song et al. (2021) [40] to classify the EEG signals. Compared to traditional ML, traditional ML assumes the data points are dispersed independently and identically. However, in many cases, the acquisition of these data is related to a subject’s physiological state (e.g., cognitive state, neural learning, language learning, emotion, electrocardiogram (ECG)). The artificial neural network-based approaches usually have better flexibility to adapt themselves in the learning of sequential data [55]. There are three salient features in the use of TSTN. First, the TSTN utilizes CSP to create discriminable features by designing a set of spatial filters. Since distinct brain areas are responsible for different brain functions [56], the use of CSP can create a specific spatial filter to extract brain activities induced from a particular task. Second, the TSTN utilizes self-attention to enhance the feature-channel data extracted from CSP. In contrast to other popular approaches using convolutional neural networks (CNNs) [45,46], the classification of CNN is related to the selection of kernel, in which large convolutional kernels are used to capture high-resolution details and small convolutional kernels are used to extract low-resolution features. The TSTN does not have to decide the kernel size. It applies self-attention on the feature-channel data in order to weight those channels which are relevant to the performing of a subject’s task. Moreover, the CNN does not consider the dependency of time-series information. Third, the TSTN slices the enhanced feature-channel data into signal patches and uses the multi-head transforming to perceive the dependencies along the temporal dimension. Compared to the previous literature using recurrent neural network (RNN) or long short-term memory networks (LSTMs) [57], both RNN and LSTM have the problems of vanishing gradients [58], so that RNN or LSTM-based solutions might have problem of dealing with EEG data induced from an MI task with a longer execution time. For example, stroke patients have slower information processing in moving the affected hand and require a longer execution time for performing mental tasks [59]. One noteworthy disadvantage of the transformer-based network is its huge model size. The TSTN requires considerable computing power and training time to achieve good accuracy. Implementation of the proposed model on wearable devices might be difficult.

In order to show the model interpretability of the TSTN neural network, the correctly classified trials were used to improve the classifiers in the continual learning processes. The relative power change of each correctly classified trial was calculated, in which the relative event-related power change was calculated using the ERD/ERS technique, formulated by Pfurtscheller and Lopes da Silva (1999) [60]. The ERD/ERS describes that the power decrease/increase within a specific frequency band anchored to a given event is calculated relative to the power of a reference period. The ERD/ERS represent the percentages of power changes according to the reference period, in which the relative power change can be represented as follows:(11)RP (%)=(A−R)/R × 100%,
where RP% is the percentage of a relative power change within alpha band (8~13 Hz) [61], A is the power of event-induced signal power, and R is the power in the reference period.

In Equation (11), the A is calculated from the period of the AO + MI or MI time block, and the reference period R is calculated from the resting period, from −2 s to 0 s, preceding the action of a virtual character or the presence of the visual cue for imagery movement. Figure 7 shows the RPs obtained from the average of the five subjects in the four datasets. In the MI and MI-FB datasets, only the RPs of the correctly classified trials were calculated. The RP of the AO + MI tasks in viewing the left-hand and right-hand movements are shown in Figure 7a and Figure 7b, respectively. The RP changes in left-hand and right-hand MI tasks are shown in Figure 7c and Figure 7d, respectively. The RP changes of the first MI-FB, second MI-FB, and third MI-FB datasets in the left-hand and right-hand movements are shown in Figure 7e and Figure 7f, Figure 7g and Figure 7h, and Figure 7i and Figure 7j, respectively. It can be observed that the ERD (the blue area of each plot in Figure 7) was enhanced with the continual learning process. This demonstrated that the TSTN network can effectively pick up those EEG trials with prominent ERDs. In Figure 7, only the ERD within the alpha band was demonstrated, because the ERD in the alpha band has been reported as the most indicative parameter in operating an MI-BCI system [61].

The BCI performance of our TSTN network was improved with the continual learning process; not only was the detection accuracy improved (see Table 1 and Table 2), but the value of ERD was also enhanced (see Figure 7). The implication between ERD and the classification accuracy echoes the findings in the previous literature that ERD is an important parameter in MI-BCI training [61]. Nevertheless, we did not see the phenomenon of ERD lateralization as mentioned in previous MI studies [62]. This might have been due to the difference in experiment design between our study and the other previous literature. In our study, we requested subjects to perform AO + MI and recall the action imagination in the following MI and MI-FB tasks. The memory recall of action imagery in MI and MI-FB tasks could involve the mirror neuron system, which presents the activations in the bilateral primary motor cortex, the primary somatosensory cortex, and the middle frontal cortex [51]. According to the study proposed by Rizzolatti et al. (2010) [63], the bilaterally distributed parietofrontal network in the mirror neuron system serves as a neural substrate to achieve the transformation of visual information into motor execution (i.e., visuomotor transformation).

Previous BCI studies only gave subjects a simple visual or auditory cue (e.g., visual or auditory cues, etc.) to perform imagery movements [64,65]. Unclear instruction could make it difficult for subjects to follow the experimenter’s guidance to achieve the desired task [65]. Since the AO task has been reported as an effective way to activate the human motor cortex with brain wave patterns similar to those induced by the MI task, we found it was easier for subjects to understand the experimenter’s instruction by requesting them to watch and follow the virtual character’s actions, instead of just providing them with simple textual instruction or visual/auditory cues. To the best of our knowledge, our research is the first study to train the BCI classifiers using EEG signals induced from AO + MI and MI tasks. Current BCI databases (e.g., BNCI Horizon; http://bnci-horizon-2020.eu/database/data-sets, accessed on 12 October 2022) do not collect both the EEG signals of AO + MI and MI tasks for the same subjects. The collected EEG data in this study could inspire future research on studying the relevance between observation and imagery movements.

One limitation of our current study is the small sample size. Because we wanted to demonstrate the training transition of a BCI classifier from AO + MI to MI-FB tasks, the subject had to join one experiment in one week and it took four weeks to complete the data collection for a subject. The small amount of data was not sufficient to have an effective comparison for the detection performances among different classifiers (see Table 3). Nevertheless, this paper aimed to study the feasibility of continual learning in a BCI classifier, from AO + MI data to MI-FB data. We observed that the detection accuracies were improved in all classifiers, and all the deep learning classifiers (i.e., TSTN, EEGNet, and DeepConvNet) showed superior performances than those in the use of traditional SVMs. The second limitation of this study is the huge computation load of our transformer framework. The transformer-based classifier has a large model size which has difficulties in coping with fast updating or fluctuation conditions.

## 5. Conclusions

In this study, we designed a BCI training method in the virtual reality (VR) environment. Subjects wore a head-mounted device (HMD) [32] and started MI training from the AO + MI task in VR environments. Unlike other MI-BCI studies which asked naïve subjects to perform unfamiliar MI tasks, the training procedure provided an easier way for subjects to conform their imagery actions. The AO-oriented training procedure also provided a more flexible design for BCI training. Subjects were only requested to follow the movements of the virtual characters and simultaneously performed imagery actions. The utilization of the AO + MI data to build the initial classifier model had several advantages. First, the instruction of AO + MI was much easier for subjects to follow [36] compared to the traditional MI task which prompts subjects to perform imagery actions by providing a simple visual/auditory cue. Second, the AO + MI provided a convenient and standardized experimental protocol. Third, the AO + MI elicited increased neural activities in motor-related brain areas, relative to the use of AO or MI only [38]. Fourth, the AO + MI has been proved as an effective way for motor learning and rehabilitation in clinics [66]. This study has answered the following issues: (1) the effectiveness of using AO + MI data to build an initial model for MI classification was validated; (2) the use of a continual learning process for the improvement of classifier performance was demonstrated; (3) the feasibility of a transformer-based deep learning model in MI-BCI classification was demonstrated; (4) the interpretability of the proposed TSTN model was shown in the analysis of alpha ERD in different BCI training steps. Our current study achieved a mean detection accuracy of 77% over the five participants in the three-class classification. In future applications, experimenters will be able to change the actions of the virtual character to train wanted imagery actions, which is important for the use of BCI in metaverse applications.

## Figures and Tables

**Figure 1 bioengineering-10-00186-f001:**
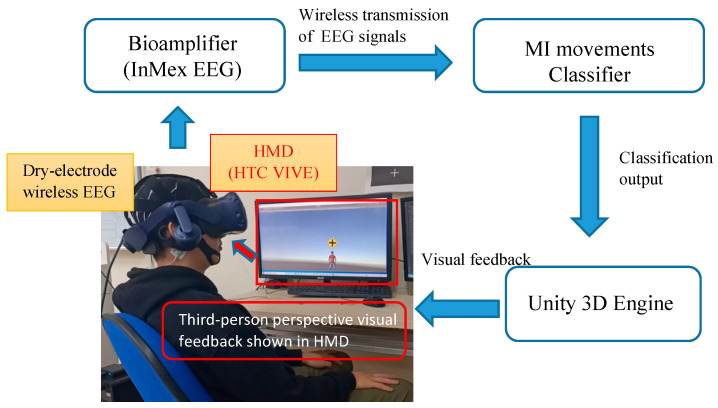
The system architecture of the proposed BCI system with immersive VR feedback.

**Figure 2 bioengineering-10-00186-f002:**
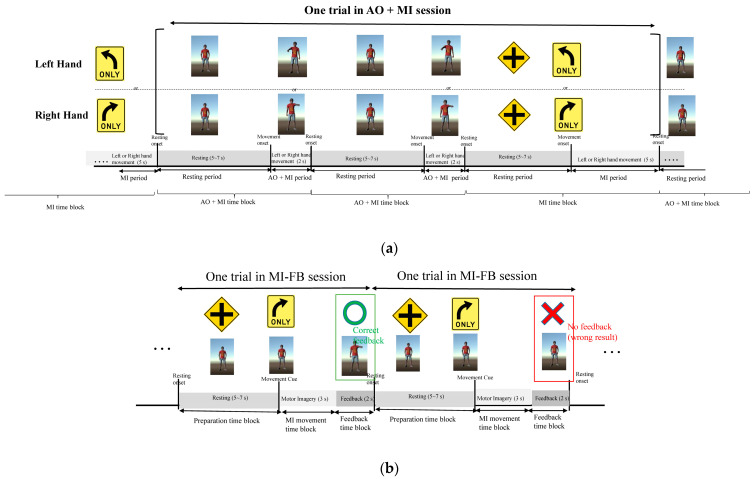
The experimental paradigms of the AO + MI and the MI − FB sessions. Subjects were asked to imagine they were looking into a mirror and the virtual character was the reflection image of themselves. (**a**) The paradigm of the AO + MI session, including two AO + MI time blocks and one MI time block in each trial; (**b**) the paradigm of the MI session, including one preparation time block, one MI movement time block, and one feedback time block.

**Figure 3 bioengineering-10-00186-f003:**
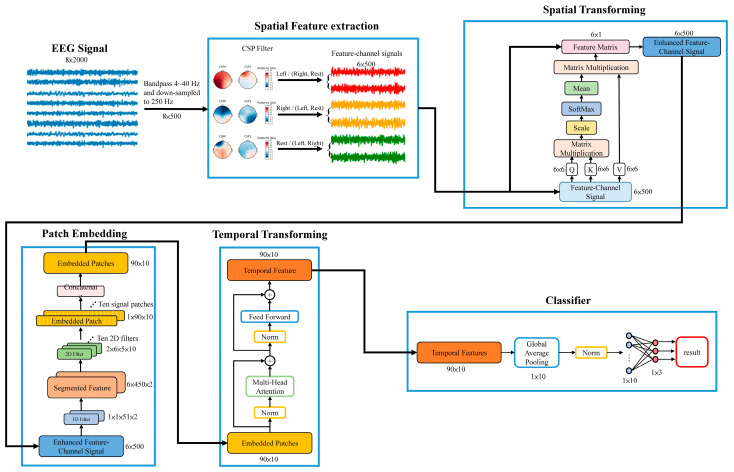
The architecture of the TSTN network. The EEG signals were prefiltered within 4~40 Hz and then downsampled to 250 Hz. The feature-channel signals were extracted using CSP, and then further enhanced by means of applying spatial transforming with an attention mechanism. The enhanced feature-channel data were segmented into ten embedded patches and the relationships among different temporal patches were perceived using multi-head transforming to obtain distinguishable representations.

**Figure 4 bioengineering-10-00186-f004:**
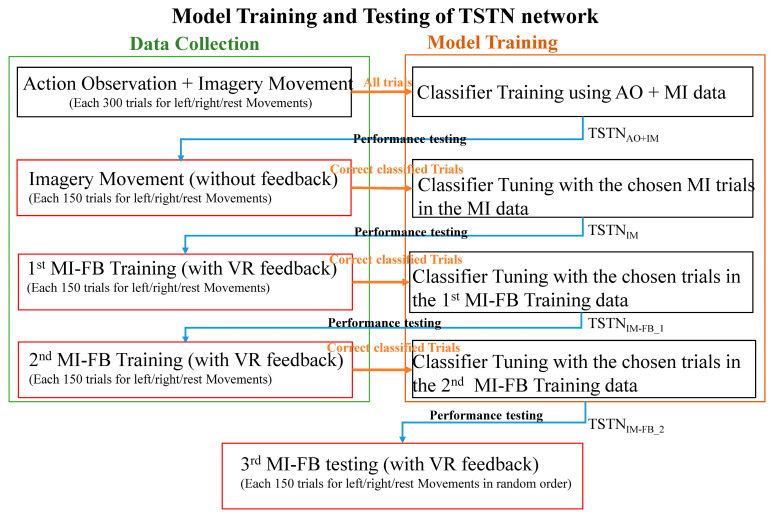
The training and testing procedure for the AO + MI, MI, 1st MI-FB, 2nd MI-FB, and 3rd MI-FB datasets.

**Figure 5 bioengineering-10-00186-f005:**
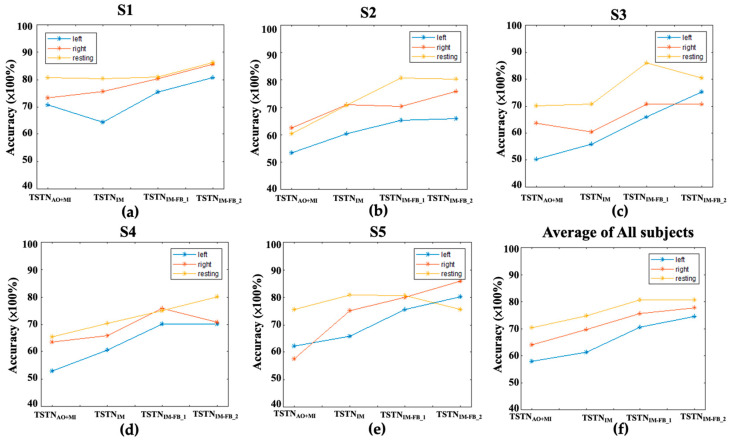
The detected accuracies by applying the TSTN_AO_, TSTN_MI_, TSTN_MI-FB_1_, and TSTN_MI-FB_2_ to the MI data (without feedback), first MI-FB data, second MI-FB data, and third MI-FB data in (**a**) S1, (**b**) S2, (**c**) S3, (**d**) S4, (**e**) S5, and (**f**) the average of all the five subjects.

**Figure 6 bioengineering-10-00186-f006:**
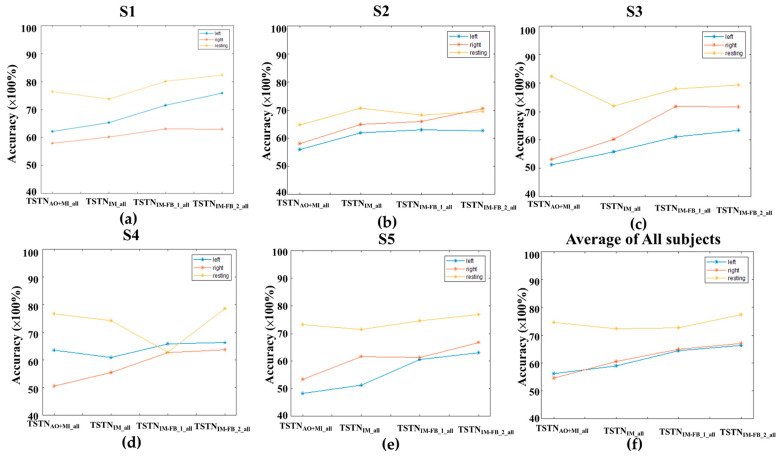
The detected accuracies by applying the TSTN_AO+MI_all_, TSTN_MI_all_, TSTN_MI-FB_1_all_, and TSTN_MI-FB_2_all_ to the MI (without feedback), first MI-FB, second MI-FB, and third MI-FB datasets in (**a**) S1, (**b**) S2, (**c**) S3, (**d**) S4, (**e**) S5 and (**f**) the average of all the five subjects.

**Figure 7 bioengineering-10-00186-f007:**
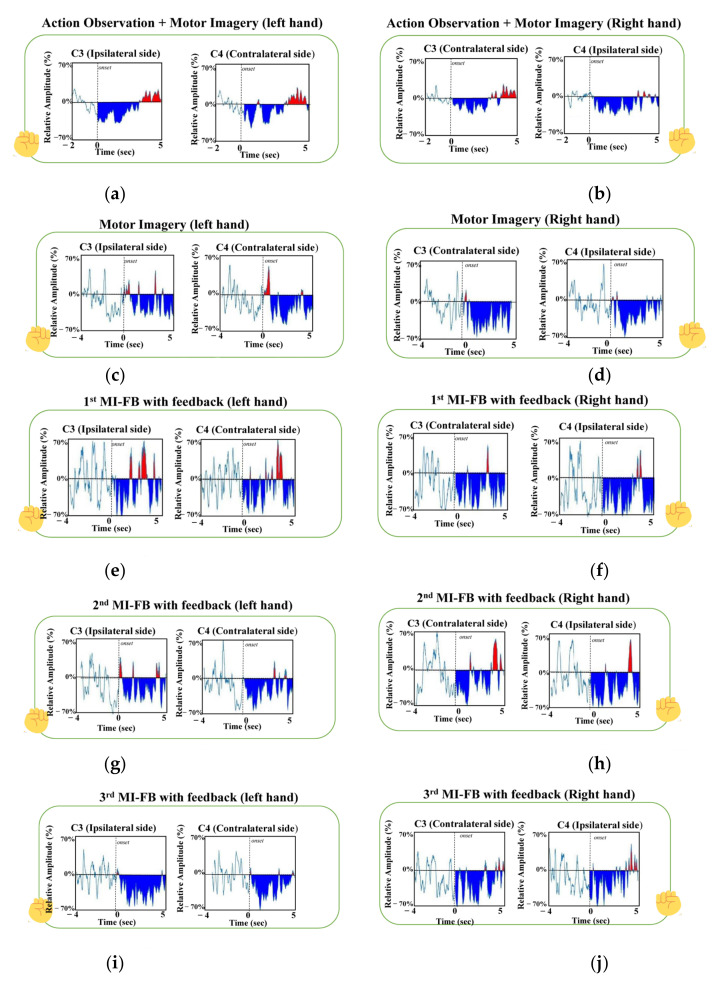
The RPs obtained from the average of the five subjects in the four datasets. The ERD and ERS are marked in blue and red colors, respectively. (**a**) The RP changes in the left-hand AO + MI task. (**b**) The RP changes in the right-hand AO + MI task. (**c**) The RP changes in the left-hand MI task. (**d**) The RP changes in the right-hand MI task. (**e**) The RP changes of the left-hand imagery movement in the first MI-FB task. (**f**) The RP changes of right-hand imagery movement in the first MI-FB task. (**g**) The RP changes of left-hand imagery movement in the second MI-FB task. (**h**) The RP changes of right-hand imagery movement in the second MI-FB task. (**i**) The RP changes of left-hand imagery movement in the third MI-FB task. (**j**) The RP changes of right-hand imagery movement in the third MI-FB task.

**Table 1 bioengineering-10-00186-t001:** Classification results of the TSTN network generated from individual data.

Classifier (Test Target)/ Subject	TSTN_AO+MI_ (IM Data)	TSTN_MI_ (1st IM-FB Data)	TSTN_MI-FB_1_ (2nd IM-FB Data)	TSTN_MI-FB_2_ (3rd IM-FB Data)
	Acc/Spec/F1	Acc/Spec/F1	Acc/Spec/F1	Acc/Spec/F1
**S1**	0.73/0.88/0.73	0.73/0.87/0.74	0.78/0.90/0.78	0.83/0.92/0.83
**S2**	0.58/0.79/0.59	0.66/0.83/0.67	0.72/0.87/0.72	0.73/0.87/0.73
**S3**	0.61/0.80/0.61	0.62/0.87/0.62	0.73/0.87/0.73	0.75/0.89/0.75
**S4**	0.61/0.80/0.61	0.65/0.83/0.65	0.73/0.84/0.74	0.73/0.87/0.74
**S5**	0.60/0.80/0.61	0.73/0.87/0.74	0.78/0.89/0.78	0.80/0.90/0.80
**Averaged accuracy**	0.63/0.81/0.63	0.68/0.84/0.68	0.75/0.87/0.75	0.77/0.89/0.77

**%Remark:** Acc: Accuracy; Spec: Specificity; F1: F1 score.

**Table 2 bioengineering-10-00186-t002:** Classification results of the TSTN network using the training data pooled for all subjects.

Classifier (Test Target)/ Subject	TSTN_AO+IM_All_ (MI Data)	TSTN_IM_All_ (1st MI-FB)	TSTN_IM-FB_1_All_ (2nd MI-FB)	TSTN_IM-FB_2_All_ (3rd MI-FB)
	Acc/Spec/F1	Acc/Spec/F1	Acc/Spec/F1	Acc/Spec/F1
**S1**	0.65/0.77/0.64	0.660.83/0.67	0.71/0.85/0.71	0.73/0.85/0.73
**S2**	0.59/0.79/0.59	0.65/0.82/0.66	0.65/0.83/0.65	0.67/0.83/0.67
**S3**	0.62/0.78/0.62	0.62/0.83/0.62	0.70/0.85/0.70	0.71/0.85/0.71
**S4**	0.63/0.81/0.63	0.63/0.81/0.63	0.65/0.84/0.65	0.69/0.83/0.69
**S5**	0.58/79/0.58	0.61/0.81/0.62	0.65/0.83/0.65	0.68/0.84/0.68
**Averaged accuracy**	0.61/0.78/0.61	0.63/0.82/0.64	0.67/0.84/0.67	0.70/0.84/0.70

**Table 3 bioengineering-10-00186-t003:** The comparisons of the detected accuracies between TSTN and SVMs with different kernel functions.

Classifier/ Test Task	TSTN	SVM_linear_	SVM_poly_	SVM_RBF_	EEGNet [44]	DeepConvNet [45]
**MI**	**0.63**	0.55	0.48	0.52	0.53	0.58
**1st MI-FB**	**0.68**	0.60	0.53	0.59	0.59	0.64
**2nd MI-FB**	**0.75**	0.67	0.56	0.67	0.69	0.72
**3rd MI-FB**	**0.77**	0.68	0.59	0.70	0.74	0.75

**%Remark:** SVM_linear_: SVM with linear kernel function; SVM_poly_: SVM with polynomial kernel function; SVM_RBF_: SVM with radial basis kernel function; EEGNet: Compact convolutional network for EEG; DeepConvNet: Deep learning with convolutional neural networks.

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
