# Peer review of "Continual Learning of a Transformer-Based Deep Learning Classifier Using an Initial Model from Action Observation EEG Data to Online Motor Imagery Classification"

_bioengineering, 2023, doi:10.3390/bioengineering10020186_

Round 1
Reviewer 1 Report
The manuscript present detection of MI activities using EEG and VR. The manuscript is presented nicely but require strict modifications. The comments are as follows:
1. In the abstract authors must present available challenges between available literature and then introduce the topic. A very little or just a line is written about method. The method must be described in brief telling about clear novelty.
2. Introduction is nicely written but require se work. Try to group into different subsection targetting the principles used.
3. More recent papers must be used by the authors. Explore some more recent papers within the journal or publisher's scope. You can have some examples as
https://www.mdpi.com/1424-8220/22/21/8128
https://www.sciencedirect.com/science/article/abs/pii/S0169260720315558
3. Try to specify the method of validation. It is unclear from the current version whether authors used hold out, crossfold or LOSO.
4. Performance measure can also be added along with the ROC and AUC curve.
5. Add the contributions, merits and demerits of the proposed model. How the proposed model overcome the limitations of the existing SoTA techniques.
6. Can the number of subjects be increased. It is only 5 subjects which are very less.
7. What are the gaps in the current study and what is the future work required to be done? This will help readers or researcher to work on the available challenges.
Author Response
Replies to the comments of Reviewer #1
(Q1) In the abstract authors must present available challenges between available literature and then introduce the topic. A very little or just a line is written about method. The method must be described in brief telling about clear novelty.
Point appreciated and modification done in the manuscript
Authors appreciate reviewer’s valuable comment. We have modified our descriptions in the abstract section to briefly describe the novelty and rationale of our method. The modification is listed as follows:
" This study adopted a transformer- based spatial-temporal network (TSTN) to decode user’s MI intentions. In contrast to other convolutional neural network (CNN) or recurrent neural network (RNN) approaches, the TSTN extracts spatial and temporal features, and applies attention mechanisms along spatial and temporal dimensions to perceive the global dependencies. The mean detection accuracies of TSTN were 0.63, 0.68, 0.75, and 0.77 in the MI, 1st MI-FB, 2nd MI-FB, and 3rd MI-FB sessions, respectively. This study demonstrated the AO+MI gave an easier way for subjects to conform their imagery actions, and the BCI performance was improved with the continual learning of MI-FB training process.”.
(Q2) Introduction is nicely written but require some work. Try to group into different subsection targetting the principles used.
Point explained and modification done in the manuscript
Authors appreciate reviewer’s comment. We divided the introduction in five parts, including: (A) the general introduction of BCI; (B) the rationale to choose motor imagery based BCI (MI-BCI) in this study; (C) the importance of visual feedback for an MI-BCI; (D) the role of VR environment in the help of BCI training; (E) the idea and objective of our study. We have checked and shortened the length of our introduction section.
(Q3) More recent papers must be used by the authors. Explore some more recent papers within the journal or publisher's scope. You can have some examples as
https://www.mdpi.com/1424-8220/22/21/8128
https://www.sciencedirect.com/science/article/abs/pii/S0169260720315558
Point appreciated and modification done in Lines 79-84.
Authors appreciate reviewer’s comment. We have added the suggested papers in the revised manuscript. The modification is listed as follows:
“ Khare et al. (2022) developed automatically tuning algorithm to optimize the selection of tunable Q wavelet transform (TQWT) parameters and high detection accuracy was achieved [17]. Khare et al. (2020) utilized flexible variational mode decomposition (F-VMD) to extract meaningful EEG components. They calculated hjorth, entropy and quartile features from F-VMD decomposed components and a flexible extreme learning machine (F-ELM) was used to classify the EEG features from different MI tasks. [18]”.
(Q4) Try to specify the method of validation. It is unclear from the current version whether authors used hold out, crossfold or LOSO.
Point appreciate and modification done in section 2.4, Line 365-370.
Authors appreciate reviewer’s comment. We have mentioned the parameter setting and training details in the section “2.4 Training of the TSTN classifier”. The modification is listed as follows:
“For the model training, Adam with a learning rate of 0.0002 was utilized and batch size was set as 50. The dropout rate was 0.3 in spatial transforming and 0.5 in temporal transforming. [40, 44] Ten-fold cross validation was applied to evaluate the final results, with each fit being performed on a training set consisting of 90% of the total training set selected at random, with the remaining 10% used as a hold out set for validation.”.
(Q5) Performance measure can also be added along with the ROC and AUC curve.
Point appreciated and modification done in Table1, Tabel2, Line 365-370.
Authors appreciate reviewer’s comment. We have added the values of specificity and F1 score in Table 1 and Table 2, in order to evaluate the adequacy of our TSTN model. The specificity shows the prediction correctness and the F1 score presents the harmonic mean of precision and recall. The modifications are listed as follows:
“Table 1 shows the classification results of using TSTN in our five participants. The test targets for the TSTNAO+MI, TSTNMI, TSTNMI-FB_1, and TSTNMI-FB_2 were the MI (without feedback), 1st MI-FB, 2nd MI-FB, and 3rd MI-FB datasets, respectively. The classification accuracies were increased after each continual learning step, in which the classification accuracies averaged over the five participants were 0.63, 0.68, 0.75, and 0.77 for the TSTNAO+MI, TSTNMI, TSTNMI-FB_1, and TSTNMI-FB_2, respectively. The specificities were 0.81, 0.84, 0.87, and 0.89 for the TSTNAO+MI, TSTNMI, TSTNMI-FB_1, and TSTNMI-FB_2, respectively. The F1 scores averaged over the five subjects were 0.63, 0.68, 0.75, and 0.77, respectively. It is worthy to note that the initial TSTNAO+MI had already achieved accuracy higher than 60% and the detection accuracies kept increased with the classifier tuning in each continual learning step. It demonstrated the feasibility of using AO+MI data to create an initial classifier model for MI classification.”.
“In addition to the model built from individual data, we are interested in testing whether the data collected from different individuals can be pooled to train a BCI classifier. The TSTNAO+MI_all, TSTNMI_all, TSTNMI-FB_1_all and the TSTNMI-FB_2_all were obtained, following the training procedure described in Fig. 4, in while the measured data over the five subjects were pooled for classifier training. The classification accuracies, specificities and F1 scores in the five subjects were listed in Table 2. The averaged accuracies in the five subjects were 0.61, 0.63, 0.68, 0.70 for the TSTNAO+MI_all, TSTNMI_all, TSTNMI-FB_1_all, and the TSTNMI-FB_2_all, respectively. The averaged specificities in the five subjects were 0.78, 0.82, 0.84, and 0.84 for the TSTNAO+MI_all, TSTNMI_all, TSTNMI-FB_1_all, and the TSTNMI-FB_2_all, re-spectively. The averaged F1 scores in the five subjects were 0.61, 0.64, 0.67, and 0.70, respectively.”.
(Q6) Add the contributions, merits and demerits of the proposed model. How the proposed model overcome the limitations of the existing SoTA techniques
Point appreciated and modification done in Line 439-454 and Line 498-454.
Authors appreciate reviewer’s comment. We have added a paragraph to describe the merit of TSTN network and its novelty compared to current CNN-based networks. The modification is listed as follows:
“To demonstrate the effectiveness of the TSTN network in this study, the EEGNet [45], DeepConvNet [46] and three SVMs with kernel functions of linear, polynomial and radial basis functions were compared. All the classifiers were built according to the training procedure listed in Fig. 4. The detected accuracies obtained from TSTN network were compared with the five classifiers and the detection accuracies were listed in Table 3. It can be observed that all the classifiers showed an increased detection accuracies in our con-tinual learning process. The three deep learning networks (i.e., the TSTN, EEGNet and DeepConvNet) showed comparable detection accuracies in the 3rd MI-FB data, which were 0.77, 0.74 and 0.75 for TSTN, EEGNet and DeepConvNet, respectively. For the initial model, the TSTNAO+MI had the highest detection accuracy (0.63) in the detection of MI data, compared to other classifiers. This might be owing to the substantial differences between the AO+MI and MI/MI-FB datasets. The AO+MI task activated both the neural circuitries of action observation and motor imagery, while the MI/MI-FB activated the neural cir-cuitry of motor imagery only. Because the transformer has the greater ability to model long-distance dependencies among temporal embedded patches, it could provide more possibilities to find the dependency between different datasets.”.
“In this paper, we adopted the transformer-based deep learning network proposed by Song et al. (2021) [40] to classify the EEG signals. Compared to traditional ML, traditional ML assumes the data points are dispersed independently and identically. However, in many cases, the acquisition of these data is related to subject’s physiological states (e.g., cognitive state, neural learning, language learning, emotion, electrocardiogram (ECG)). The artificial neural network based approaches usually have better flexibility to adapt themselves in the learning of sequential data [55]. There are three salient features in the use of TSTN. First, the TSTN utilizes CSP to create discriminable features by designing a set of spatial filters. Since distinct brain areas are responsible for different brain functions [56], the use of CSP can create a specific spatial filter to extract brain activities induced from a particular task. Second, the TSTN utilizes self-attention to enhance the feature-channel data extracted from CSP. In contrast to other popular approaches using convolutional neural networks (CNNs) [45, 46], the classification of CNN is related to the selection of kernel, in which large convolutional kernels are used to capture high-resolution details and small convolutional kernels are used to extract low-resolution features. The TSTN doesn’t have to decide the kernel size. It applies self-attention on the feature-channel data in order to weight those channels which are relevant to the performing of subject’s task. Besides, the CNN doesn’t consider the dependency of time-series information. Third, the TSTN slices the enhanced feature-channel data into signal patches and uses the multi-head transforming to perceive the dependencies along temporal dimension. Compared to previous literatures using recurrent neural network (RNN) or long short-term memory networks (LSTMs) [57], both RNN and LSTM have the problems of vanishing gradients [58], so that RNN or LSTM based solutions might have problem to deal with EEG data induced from MI task with longer execution time. For example, stroke patients have slower information processing in moving affected hand and require longer execution time for performing mental tasks [59]. One noteworthy disadvantage of transformer-based network is its huge model size. The TSTN requires considerable computing power and training time to achieve good accuracy. Implementation of the proposed model on wearable devices might be difficult.”.
(Q7) Can the number of subjects be increased. It is only 5 subjects which are very less.
Point appreciate and modification done in Line 582-594.
In this study, we want to demonstrate the training transition of a BCI classifier from AO+MI to MI-FB tasks. Subjects had to join one experiment in one week and it cost four weeks to complete the data collection in a subject. The feature of this study is that this is the study to have a complete set of AO+MI, MI, and three MI-FB datasets in one subject. That is why the data acquisition in this study is valuable and the subject number is few.
The comparison of detection performance among different classifiers is not the main purpose of this study. We can understand the small amount of data is not sufficient to have an effective comparison for the detection performances among different classifiers. Therefore, we have removed the statistical comparison of detection performance in Table 3. In order to clarify this point, we have added one paragraph to describe the limitation of the few subject number in this study. The modification is listed as follows:
“One limitation of our current study is the small sample size. Because we want to demonstrate the training transition of a BCI classifier from AO+MI to MI-FB tasks, subject had to join one experiment in one week and it cost four weeks to complete the data col-lection in a subject. The small amount of data is not sufficient to have an effective com-parison for the detection performances among different classifiers (see Table 3). Never-theless, this paper aimed to study the feasibility of continually learning in a BCI classifier, from AO+MI data to MI-FB data. We observed that the detection accuracies were improved in all classifiers, and all the deep learning classifiers (i.e., TSTN, EEGNet and Deep-ConvNet) showed superior performances than those in the use of traditional SVMs. The second limitation of this study is the huge computation load of our transformer framework. The transformer-based classifier has large model size which has difficulties in coping with fast updating or fluctuation conditions.”.
(Q8) What are the gaps in the current study and what is the future work required to be done? This will help readers or researcher to work on the available challenges.
Point appreciate and modification done in conclusion section, Line 596-602 and Line 616-618.
Authors appreciate reviewer’s comment. In this study, we designed an BCI training method in virtual reality (VR) environment. Subjects wore head-mounted device (HMD) and started MI training from AO+MI task in VR environments. Unlike other MI-BCI studies which asked naïve subjects to perform unfamiliar MI tasks, the training procedure provides an easier way for subjects to conform their imagery actions. The AO-oriented training procedure also provided a more flexible design for BCI training.
We have described the problem in current BCI studies and the applicability of this AO-oriented BCI training for metaverse application in future studies.
The modifications are listed below:
“In this study, we designed an BCI training method in virtual reality (VR) envi-ronment. Subjects wore head-mounted device (HMD) [32] and started MI training from AO+MI task in VR environments. Unlike other MI-BCI studies which asked naïve subjects to perform unfamiliar MI tasks, the training procedure provides an easier way for subjects to conform their imagery actions. The AO-oriented training procedure also provided a more flexible design for BCI training. Subjects were just requested to follow the move-ments of the virtual characters and simultaneously performed imagery actions.”
“In future applications, experimenters are able to change the actions of the virtual character to train wanted imagery actions, which is important for the use of BCI in metaverse applications.”.

Reviewer 2 Report
This is an interesting manuscript on continual learning of a transformer-based deep learning classifier. Authors emply an intial model from action observation EEG data to online motor imagery classification.
I am very positive towards this work but I have two minor points towards the methodology:
-What reference electrodes have you used?
-More information on the noise treatment
Minor:
-imagery movement (MI) does no correspond to the capital letters described in the text
Plese, review typos (e.g., extra spaces during the text)
Author Response
Replies to the comments of Reviewer #2
(Q1) What reference electrodes have you used?
Point appreciate and modification done in Line 185-187.
Authors appreciate reviewer’s comment. Our EEG measurement followed international 10-20 system with a reference electrode placed at right mastoid and a ground electrode placed at left mastoid, in which monopolar measurement montage was applied. We have added the reference and ground electrode placement in the manuscript. The modification is listed as follows:
“The EEG electrodes were placed in accordance with international 10-20 system. Electrodes were located at FC1, FC2, C3, Cz, C4, P1 and P2 positions, with respect to a reference electrode placed at right mastoid and a ground electrode placed at left mastoid.”.
(Q2) More information on the noise treatment.
Point appreciated and modification done in Line 253-256.
The EEG signals were sampled at 1kHz and pre-filtered within 4~40Hz. The filtered signals were downsampled to 250Hz as input data for TSTN network. The description in section 2.3 is listed as follows:
“In this study, we adopted the work proposed by Song et al. (2021) [33], which applied attention mechanisms on the spatial and temporal features of EEG data. The eight-channel EEG signals were prefiltered within 4~40Hz (3rd-order Butterworth IIR filter) and then downsampled to 250Hz. The EEG data were then segmented into two-second epochs….”.
(Q3) imagery movement (MI) does no correspond to the capital letters described in the text
Point appreciate and modification done in the manuscript.
Author appreciate reviewer’s comment. We have changed the term “imagery movement” to “motor imagery”, in order to coherent with the abbreviation term “motor imagery (MI)”.
(Q4) Please, review typos (e.g., extra spaces during the text)
Point appreciate and modification done in the manuscript
Authors appreciate reviewer’s comment. We have checked the manuscript carefully and corrected all the typos.

Reviewer 3 Report
Reviewer’s Report on the manuscript entitled:
Continual Learning of a transformer-based deep learning classifier using an initial model from Action Observation EEG data to online motor imagery classification
The authors purposed a brain computer interface training procedure in virtual reality environment for stroke rehabilitation purposes. Though the topic and results are interesting, the presentation is poor, and there are many grammar issues. Furthermore, the test of 5 subjects is not enough in my view. Please see below my comments.
The title has a typo: “initial” not “intial”
The abstract must be improved. The motivation must be given first, then the objective and method, and results and recommendation need to be mentioned concisely. Please also avoid using too many acronyms in the abstract.
Line 13. MI is not a suitable abbreviation for Closed-loop imagery movement.
Line 14. What is BCI? Please define all the abbreviations the first time they appear both in abstract and body of the manuscript.
Line 24. “an” easier way. Grammar issue.
Lines 40-42. Please also add a few sentences to describe the EEG, noise source, such as eye-blink or muscle movement, applications, such as EEG emotion classification, seizure detection, sleep disorder, etc. Some suggested articles that can be included here are:
https://doi.org/10.3390/s22082948
https://doi.org/10.3390/signals1010003
https://doi.org/10.3390/s22062346
Line 52. Please remove “however”. Grammar issue. Similarly in line 92, etc.
Line 118. “to train” not “to trained”
Line 149. “can provide” not “can provides”
Line 551. Please elaborate more what the objective of this research is.
Line 166. “an easier way”
The Introduction is long and can be shortened a bit. You may also consider a subsection called “Literature Review” or “Previous Studies”.
Figure 3. Some of the texts in the boxes or arrows are not clearly written. For example: Filt 4~ 40Hz down sampling, “seg ment” etc. Please fix them.
Figure 5. Please insert all the values on the y-axis (accuracy).
Lines 417 to 419. This can be shortened as
"The detection accuracies of the four classifiers in (a) S1, (b) S2, …, (e) S5."
Table 3. Is this average accuracy 0.71 acceptable? I think selecting 5 subjects in this research is not adequate to perform a rigorous statistical testing. Can authors include more subjects (different gender and age group)? Like Alimardani et al. (2022) [13] where they tested 54 subjects. That number is more reliable to draw a conclusion about the overall accuracy of your models.
Line 468. Grammar issue. Please rewrite.
Line 551. Please use only past verbs in Conclusions. For example. “The purpose of this research was to…”
Regards,
Author Response
Replies to the comments of Reviewer #3
(Q1) The title has a typo: “initial” not “intial”
Point appreciate and modification done in the manuscript.
We have corrected the typo in our title.
(Q2) The abstract must be improved. The motivation must be given first, then the objective and method, and results and recommendation need to be mentioned concisely. Please also avoid using too many acronyms in the abstract.
Point appreciate and modification done in the manuscript.
Authors appreciate reviewer’s comment. We have rewritten our abstract to comply with reviewer’s valuable suggestion. We have highlighted the objective in the beginning and the followed the structure of suggested by reviewer. The modifications are listed as follows:
“ The motor imagery (MI)-based brain computer interface (BCI) is an intuitive interface that enables users to communicate with external environments through their minds. However, current MI-BCI systems ask naïve subjects to perform unfamiliar MI tasks with simple textual instruction or vis-ual/auditory cue. The unclear instruction for MI execution not only results in large inter-subject variability in the measured EEG patterns but also causes the difficulty of grouping cross-subject data for big-data training. In this study, we designed an BCI training method in virtual reality (VR) environment. Subjects wore head-mounted device (HMD) and executed action observation (AO) concurrently with MI (i.e., AO+MI) in VR environments. EEG signals recorded in AO+MI task were used to train an initial model, and the initial model was continually improved by the provision of EEG data in the following BCI training sessions. We recruited five healthy subjects and each subject was requested to participate in three kinds of tasks, including an AO+MI task, an MI task, and the task of MI with visual feedback (MI-FB) for three times. This study adopted a transformer- based spatial-temporal network (TSTN) to decode user’s MI intentions. In contrast to other convolutional neural network (CNN) or recurrent neural network (RNN) approaches, the TSTN extracts spatial and temporal features, and applies attention mechanisms along spatial and temporal dimensions to perceive the global dependencies. The mean detection accuracies of TSTN were 0.63, 0.68, 0.75, and 0.77 in the MI, 1st MI-FB, 2nd MI-FB, and 3rd MI-FB sessions, re-spectively. This study demonstrated the AO+MI gave an easier way for subjects to conform their imagery actions, and the BCI performance was improved with the continual learning of MI-FB training process.”.
(Q3) Line 13. MI is not a suitable abbreviation for Closed-loop imagery movement.
Point appreciate and modification done in the abstract.
We have rewritten the sentence. The first sentence in abstract section has been changed as:
“The motor imagery (MI) based brain computer interface (BCI) is an intuitive interface that enables users to communicate with external environments through their minds….”.
(Q4) Line 14. What is BCI? Please define all the abbreviations the first time they appear both in abstract and body of the manuscript.
Point appreciate and modification done in Line 36.
The term “brain computer interface” for the acronym “BCI” has been mentioned in the first sentence of abstract section.
(Q5) Line 24. “an” easier way. Grammar issue.
Point appreciate and modification done in the manuscript.
The description “a easier way” has been corrected as “an easier way” in the manuscript.
(Q6) Lines 40-42. Please also add a few sentences to describe the EEG, noise source, such as eye-blink or muscle movement, applications, such as EEG emotion classification, seizure detection, sleep disorder, etc. Some suggested articles that can be included here are:
https://doi.org/10.3390/s22082948
https://doi.org/10.3390/signals1010003
https://doi.org/10.3390/s22062346
Point appreciate and modification done in Line 46-52.
Authors appreciate reviewer’s comment. The descriptions about EEG’s artifact and noise removal have been added in the manuscript. The modifications are listed as follows:
“Especially, the EEG has the advantages of easy preparation, inexpensive equipment cost, and high temporal resolution. It has been chosen for a wide variety of clinical applications, such as sleep disorder diagnosis [3], seizure detection [4], emotion classification [5], etc. Advanced signal processing techniques for EEG also have been developed to avoid the interference of external artifacts (e.g., electromyography, motion artifact, etc.) [6, 7]. Owing to the aforementioned reasons, EEG is the most popular choice to implement a BCI system [7].”.
(Q7) Line 52. Please remove “however”. Grammar issue. Similarly in line 92, etc.
Point appreciate and modification done in the manuscript.
The word “however” has been removed.
(Q8) Line 118. “to train” not “to trained”
Point appreciate and modification done in the manuscript.
The word “to train” has been corrected.
(Q9) Line 149. “can provide” not “can provides”
Point appreciate and modification done in the manuscript.
The word “can provide” has been corrected.
(Q10) Line 551. Please elaborate more what the objective of this research is.
Point appreciate and modification done in the manuscript.
Authors appreciate reviewer’s comment. We have rewritten the first part of our conclusion section to clarify the objective of this research. The modifications are listed as follows:
“In this study, we designed an BCI training method in virtual reality (VR) envi-ronment. Subjects wore head-mounted device (HMD) [32] and started MI training from AO+MI task in VR environments. Unlike other MI-BCI studies which asked naïve subjects to perform unfamiliar MI tasks, the training procedure provides an easier way for subjects to conform their imagery actions. The AO-oriented training procedure also provided a more flexible design for BCI training. Subjects were just requested to follow the move-ments of the virtual characters and simultaneously performed imagery actions.”.
(Q11) Line 166. “an easier way”
Point appreciate and modification done in the manuscript.
The typo “an easier way” has been corrected.
(Q12) The Introduction is long and can be shortened a bit. You may also consider a subsection called “Literature Review” or “Previous Studies”.
Point appreciate and modification done in the manuscript.
Authors appreciate reviewer’s comment. We divided the introduction in five parts, including: (A) the general introduction of BCI; (B) the rationale to choose motor imagery based BCI (MI-BCI) in this study; (C) the importance of visual feedback for an MI-BCI; (D) the role of VR environment in the help of BCI training; (E) the idea and objective of our study. We have checked and shortened the length of our introduction section.
(Q13) Figure 3. Some of the texts in the boxes or arrows are not clearly written. For example: Filt 4~ 40Hz down sampling, “seg ment” etc. Please fix them.
Point appreciate and modification done in the manuscript.
Authors appreciate reviewer’s comment. We have checked the label of each block in Fig.3. The figure has been corrected as follows:
Figure 3. The architecture of the TSTN network. The EEG signals were prefiltered within 4~40Hz and then downsampled to 250Hz. The feature-channel signals were extracted using CSP, and then further enhanced by means of applying spatial transforming with attention mechanism. The enhanced feature-channel data were segmented into embedded patches and the relationships among different temporal patches were perceived using multi-head transforming to obtain distinguishable representations.
(Q14) Figure 5. Please insert all the values on the y-axis (accuracy).
Point appreciate and modification done in the manuscript.
Authors appreciate reviewer’s comment. We have inserted the values of y-axis in both Fig. 5 and Fig. 6.
Figure 5. The detected accuracies by applying the TSTNAO, TSTNMI, TSTNMI-FB_1, and TSTNMI-FB_2 to the MI data (without feedback), 1st MI-FB data, 2nd MI-FB data, and 3rd MI-FB data in (a) S1, (b) S2, (c) S3, (d) S4, and (e) the average of all the five subjects.
Figure 6. The detected accuracies by applying the TSTNAO+MI_all, TSTNMI_all, TSTNMI-FB_1_all, and TSTNMI-FB_2_all to the MI (without feedback), 1st MI-FB, 2nd MI-FB, and 3rd MI-FB datasets in (a) S1, (b) S2, (c) S3, (d) S4, and (e) the average of all the five subjects.
(Q15) Lines 417 to 419. This can be shortened as"The detection accuracies of the four classifiers in (a) S1, (b) S2, …, (e) S5."
Point appreciate and modification done in the manuscript.
Authors appreciate reviewer’s comment. The figure captions of Fig. 5 and Fig. 6 have been shorted which are listed as follows:
“Figure 5. The detected accuracies by applying the classifierAO, classifierMI, classifierMI-FB_1, and classifierMI-FB_2 to the MI data (without feedback), 1st MI-FB data, 2nd MI-FB data, and 3rd MI-FB data in (a) S1, (b) S2, (c) S3, (d) S4, and (e) the average of all the five subjects.”
“Figure 6. The detected accuracies by applying the classifierAO+MI_all, classifierMI_all, classifierMI-FB_1_all, and classifierMI-FB_2_all to the MI (without feedback), 1st MI-FB, 2nd MI-FB, and 3rd MI-FB datasets in (a) S1, (b) S2, (c) S3, (d) S4, and (e) the average of all the five subjects.”
(Q16) Table 3. Is this average accuracy 0.71 acceptable? I think selecting 5 subjects in this research is not adequate to perform a rigorous statistical testing. Can authors include more subjects (different gender and age group)? Like Alimardani et al. (2022) [13] where they tested 54 subjects. That number is more reliable to draw a conclusion about the overall accuracy of your models.
Point appreciate and modification done in Line 582-594.
In this study, we want to demonstrate the training transition of a BCI classifier from AO+MI to MI-FB tasks. Subjects had to join one experiment in one week and it cost four weeks to complete the data collection in a subject. The feature of this study is that this is the study to have a complete set of AO+MI, MI, and three MI-FB datasets in one subject. That is why the data acquisition in this study is valuable and the subject number is few.
The comparison of detection performance among different classifiers is not the main purpose of this study. We can understand the small amount of data is not sufficient to have an effective comparison for the detection performances among different classifiers. Therefore, we have removed the statistical comparison of detection performance in Table 3. In order to clarify this point, we have added one paragraph to describe the limitation of the few subject number in this study. The modification is listed as follows:
“One limitation of our current study is the small sample size. Because we want to demonstrate the training transition of a BCI classifier from AO+MI to MI-FB tasks, subject had to join one experiment in one week and it cost four weeks to complete the data col-lection in a subject. The small amount of data is not sufficient to have an effective com-parison for the detection performances among different classifiers (see Table 3). Never-theless, this paper aimed to study the feasibility of continually learning in a BCI classifier, from AO+MI data to MI-FB data. We observed that the detection accuracies were improved in all classifiers, and all the deep learning classifiers (i.e., TSTN, EEGNet and Deep-ConvNet) showed superior performances than those in the use of traditional SVMs. The second limitation of this study is the huge computation load of our transformer framework. The transformer-based classifier has large model size which has difficulties in coping with fast updating or fluctuation conditions.”.
(Q17) Line 468. Grammar issue. Please rewrite.
Point explained and modification done in the manuscript.
The sentence has been corrected as “The adoption of distinct MI strategies could result in large inter-individual variations in the induced brain wave patterns, which makes difficulty in the group analysis of MI data.”.
(Q18) Line 551. Please use only past verbs in Conclusions. For example. “The purpose of this research was to…”
Point explained and modification done in the manuscript.
Authors appreciate reviewer’s comment. The conclusion section has been rewritten as follows:
“In this study, we designed an BCI training method in virtual reality (VR) envi-ronment. Subjects wore head-mounted device (HMD) [32] and started MI training from AO+MI task in VR environments. Unlike other MI-BCI studies which asked naïve subjects to perform unfamiliar MI tasks, the training procedure provides an easier way for subjects to conform their imagery actions. The AO-oriented training procedure also provided a more flexible design for BCI training. Subjects were just requested to follow the move-ments of the virtual characters and simultaneously performed imagery actions. The uti-lization of the AO+MI data to build the initial classifier model had several advantages. First, the instruction of AO+MI was much easier for subjects to follow [36], compared to the traditional MI task which prompted subjects to perform imagery actions by providing a simple visual/auditory cue. Second, the AO+MI provided a convenient and standardized experimental protocol. Third, the AO+MI elicited increased neural activities in mo-tor-related brain areas, relative to the use of AO or MI only [38]. Fourth, the AO+MI has been proved as an effective way for motor learning and rehabilitation in clinics [66]. This study has answered the following issues: (1) The effectiveness of using AO+MI data to build an initial model for MI classification was validated; (2) The use of continual learning process for the improvement of classifier performance was demonstrated; (3) The feasi-bility of transformer-based deep learning model in MI-BCI classification was demon-strated; (4) The interpretability of the proposed TSTN model was shown in the analysis of alpha ERD in different BCI training steps. Our current study has achieved mean detection accuracy of 77% over the five participants in the three-class classification. In future ap-plications, experimenters are able to change the actions of the virtual character to train wanted imagery actions, which is important for the use of BCI in metaverse applications. “.

Reviewer 4 Report
This topic is very interesting, however this manuscript needs a revision. Look at these points to improve it:
- Lines 28-152. Introduction section is very long. Try to merge it. In addition, it is not clear what is the aim of this paper. "In this study, we intend to study the feasibility of building the initial classifier for 142 BCI system using the EEG data obtained from action observation (AO) task" Please improve this point.
- Lines 134-137: ". Recent studies also demonstrate the training efficacies can 134 be enhanced by creating the illusions of ownership and agency [13, 21, 23] which can be 135 conveniently achieved by wearing a VR head-mount device (HMD) [14, 19, 20]" Also these paper should be added about this topic: --- Brain Tumor and Augmented Reality: New Technologies for the Future. Int J Environ Res Public Health. 2022 May 23;19(10):6347. doi: 10.3390/ijerph19106347. --- The New Frontier: A Review of Augmented Reality. Aesthet Surg J. 2019 Aug 22;39(9):1017-1018. doi: 10.1093/asj/sjz098.
- Lines 534-549. "In our study, we requested subjects to perform AO+MI and recall 540 the action imagination in the following MI and MI-FB tasks. " What does this paper add new to the literature?
- Lines 337-338: "Figure 3. The architecture of the TSTN network." Improve this figure's legend.
- Please discuss about telemedicine that offers better timeliness of some medical treatments, with several benefits for patients, however the lack of a standard legal framework causes some doubts about patient privacy, liability coverage for treating "healthcare workers and financial reimbursements by insurance companies" ref. -- Telemedicine: Could it represent a new problem for spine surgeons to solve? Global Spine J. 2022 Jul;12(6):1306-1307. doi: 10.1177/21925682221090891.
- Lines 551-552. Conclusion section: "The purpose of this research is to build a training procedure in virtual reality (VR) 551 environment" This is not a conclusion! Revise.
- Lines 544-549. Please add a "study limitations" section. One limitation is the small sample of subjects, for example.
Author Response
Replies to the comments of Reviewer #4
(Q1) - Lines 28-152. Introduction section is very long. Try to merge it. In addition, it is not clear what is the aim of this paper. "In this study, we intend to study the feasibility of building the initial classifier for 142 BCI system using the EEG data obtained from action observation (AO) task" Please improve this point.
Point appreciate and modification done in Line 13-18, Line 125-130, and Line 596-602.
Authors appreciate reviewer’s comment. We divided the introduction in five parts, including: (A) the general introduction of BCI; (B) the rationale to choose motor imagery based BCI (MI-BCI) in this study; (C) the importance of visual feedback for an MI-BCI; (D) the role of VR environment in the help of BCI training; (E) the idea and objective of our study. We have checked and shortened the length of our introduction section.
This study considered that most current MI-BCI systems ask naïve subjects to perform unfamiliar MI tasks with simple textual in-struction or visual/auditory cue. The unclear instruction for MI execution not only results in large inter-subject variability in the measured EEG patterns but also causes the difficulty of grouping cross-subject data for big-data training. Therefore, in this study, we designed an BCI training method in virtual reality (VR) envi-ronment. Subjects wore head-mounted device (HMD) and started MI training from AO+MI task in VR environments. Unlike other MI-BCI studies which asked naïve subjects to perform unfamiliar MI tasks, the training procedure provides an easier way for subjects to conform their imagery actions. The AO-oriented training procedure also provided a more flexible design for BCI training.
(Q2)- Lines 134-137: ". Recent studies also demonstrate the training efficacies can 134 be enhanced by creating the illusions of ownership and agency [13, 21, 23] which can be conveniently achieved by wearing a VR head-mount device (HMD) [14, 19, 20]" Also these paper should be added about this topic: --- Brain Tumor and Augmented Reality: New Technologies for the Future. Int J Environ Res Public Health. 2022 May 23;19(10):6347. doi: 10.3390/ijerph19106347. --- The New Frontier: A Review of Augmented Reality. Aesthet Surg J. 2019 Aug 22;39(9):1017-1018. doi: 10.1093/asj/sjz098.
Point explained and modification done in Line 137.
Authors appreciate reviewer’s comment. The papers have been cited in the manuscript. The modification is listed as follow:
“Recent studies also demonstrate the training efficacies can be enhanced by creating the illusions of ownership and agency [20, 28, 30] which can be conveniently achieved by wearing a VR head-mount device (HMD) [21, 26, 27, 31, 32].”
(Q3) - Lines 534-549. "In our study, we requested subjects to perform AO+MI and recall the action imagination in the following MI and MI-FB tasks. " What does this paper add new to the literature?
Point explained and modification done in the abstract, introduction, discussion, and conclusion sections.
In this study, we designed an BCI training method in virtual reality (VR) environment. Subjects wore head-mounted device (HMD) and started MI training from AO+MI task in VR environments. Unlike other MI-BCI studies which asked naïve subjects to perform unfamiliar MI tasks, the training procedure provides an easier way for subjects to conform their imagery actions. The AO-oriented training procedure also provided a more flexible design for BCI training. Subjects were just requested to follow the movements of the virtual characters and simultaneously performed imagery actions. The utilization of the AO+MI data to build the initial classifier model had several advantages. First, the instruction of AO+MI was much easier for subjects to follow, compared to the traditional MI task which prompted subjects to perform imagery actions by providing a simple visual/auditory cue. Second, the AO+MI provided a convenient and standardized experimental protocol. Third, the AO+MI elicited increased neural activities in motor-related brain areas, relative to the use of AO or MI only. Fourth, the AO+MI has been proved as an effective way for motor learning and rehabilitation in clinics. This study has answered the following issues: (1) The effectiveness of using AO+MI data to build an initial model for MI classification was validated; (2) The use of continual learning process for the improvement of classifier performance was demonstrated; (3) The feasibility of transformer-based deep learning model in MI-BCI classification was demonstrated; (4) The interpretability of the proposed TSTN model was shown in the analysis of alpha ERD in different BCI training steps. Our current study has achieved mean detection accuracy of 77% over the five participants in the three-class classification. In future applications, experimenters are able to change the actions of the virtual character to train wanted imagery actions, which is important for the use of BCI in metaverse applications.
(Q4) - Lines 337-338: "Figure 3. The architecture of the TSTN network." Improve this figure's legend.
Point explained and modification done in the manuscript.
Authors appreciate reviewer’s comment. The figure legned of Fig. 3 has been improved and listed as follows:
“Figure 3. The architecture of the TSTN network. The EEG signals were prefiltered within 4~40Hz and then downsampled to 250Hz. The feature-channel signals were extracted using CSP, and then further enhanced by means of applying spatial transforming with attention mechanism. The en-hanced feature-channel data were segmented into embedded patches and the relationships among different temporal patches were perceived using multi-head transforming to obtain distinguisha-ble representations.”.
(Q5) - Please discuss about telemedicine that offers better timeliness of some medical treatments, with several benefits for patients, however the lack of a standard legal framework causes some doubts about patient privacy, liability coverage for treating "healthcare workers and financial reimbursements by insurance companies" ref. -- Telemedicine: Could it represent a new problem for spine surgeons to solve? Global Spine J. 2022 Jul;12(6):1306-1307. doi: 10.1177/21925682221090891.
Point appreciated and explained.
This study tried to propose a new training procedure for MI-based BCI, and the transformer based TSTN network was applied to discriminate the imagery movements. Because this study is not related to telemedicine, we want to have reviewer’s understanding about why we didn’t add this reference in our revised manuscript.
(Q6) - Lines 551-552. Conclusion section: "The purpose of this research is to build a training procedure in virtual reality (VR) [1]environment" This is not a conclusion! Revise.
Point appreciated and modification done in the conclusion section.
We have rewritten our conclusion section. The sentence has been removed. The corrected conclusion is listed as follows:
“In this study, we designed an BCI training method in virtual reality (VR) environment. Subjects wore head-mounted device (HMD) [32] and started MI training from AO+MI task in VR environments. Unlike other MI-BCI studies which asked naïve subjects to perform unfamiliar MI tasks, the training procedure provides an easier way for subjects to conform their imagery actions. The AO-oriented training procedure also provided a more flexible design for BCI training. Subjects were just requested to follow the move-ments of the virtual characters and simultaneously performed imagery actions.”.
(Q7)- Lines 544-549. Please add a "study limitations" section. One limitation is the small sample of subjects, for example.
Point appreciated and modification done in Line 582-593.
Authors appreciate reviewer’s comment. We have added one paragraph to describe the limitation of this study. The modification is listed as follows:
“One limitation of our current study is the small sample size. Because we want to demonstrate the training transition of a BCI classifier from AO+MI to MI-FB tasks, subject had to join one experiment in one week and it cost four weeks to complete the data col-lection in a subject. The small amount of data is not sufficient to have an effective com-parison for the detection performances among different classifiers (see Table 3). Never-theless, this paper aimed to study the feasibility of continually learning in a BCI classifier, from AO+MI data to MI-FB data. We observed that the detection accuracies were improved in all classifiers, and all the deep learning classifiers (i.e., TSTN, EEGNet and Deep-ConvNet) showed superior performances than those in the use of traditional SVMs. The second limitation of this study is the huge computation load of our transformer framework. The transformer-based classifier has large model size which has difficulties in coping with fast updating or fluctuation conditions.”.

Round 2
Reviewer 1 Report
The comments are addressed. It looks good for publication.
Reviewer 3 Report
I would like to thank the authors for addressing my comments satisfactorily.
Please carefully proofread your manuscript for any typos/punctuation issues and double-check all your figures/tables.
Regards,
Reviewer 4 Report
Authors solved all my criticisms.